# Using Routinely Collected Health Records to Identify the Fine-Resolution Spatial Patterns of Soil-Transmitted Helminth Infections in Rwanda

**DOI:** 10.3390/tropicalmed7080202

**Published:** 2022-08-22

**Authors:** Elias Nyandwi, Tom Veldkamp, Sherif Amer, Eugene Ruberanziza, Nadine Rujeni, Ireneé Umulisa

**Affiliations:** 1Centre for Geographic Information Systems and Remote Sensing, College of Science and Technology, University of Rwanda, KN 73 Street, (Avenue de l’Armée), Kigali P.O. Box 4285, Rwanda; 2Faculty of Geo-Information Science and Earth Observation, University of Twente, 7522 NB Enschede, The Netherlands; 3Rwanda Biomedical Centre, Malaria and Other Parasitic Diseases Unit, KG 644 Street, Kimihurura, Kigali P.O. Box 4285, Rwanda; 4School of Health Sciences, College of Medicine and Health Sciences, University of Rwanda, KN 73 Street, (Avenue de l’Armée), Kigali P.O. Box 4285, Rwanda

**Keywords:** spatial pattern, Rwanda, soil-transmitted helminth infection, GIS

## Abstract

Background. Soil-transmitted helminths (STH) are parasitic diseases with significant public health impact. Analysis is generally based on cross-sectional prevalence surveys; outcomes are mostly aggregated to larger spatial units. However, recent research demonstrates that infection levels and spatial patterns differ between STH species and tend to be localized. Methods. Incidence data of STHs including roundworm (*Ascaris lumbricoides*), whipworm (*Trichuris trichiura*) and hookworms per primary health facility for 2008 were linked to spatially delineated primary health center service areas. Prevalence data per district for individual and combined STH infections from the 2008 nationwide survey in Rwanda were also obtained. Results. A comparison of reported prevalence and incidence data indicated significant positive correlations for roundworm (R^2^ = 0.63) and hookworm (R^2^ = 0.27). Weak positive correlations were observed for whipworm (R^2^ = 0.02) and the three STHs combined (R^2^ = 0.10). Incidence of roundworm and whipworm were found to be focalized with significant spatial autocorrelation (Moran’s I > 0: 0.05–0.38 and *p* ≤ 0.03), with (very) high incidence rates in some focal areas. In contrast, hookworm incidence is ubiquitous and randomly distributed (Moran’s I > 0: 0.006 and *p* = 0.74) with very low incidence rates. Furthermore, an exploratory regression analysis identified relationships between helminth infection cases and potential environmental and socio-economic risk factors. Conclusions. Findings show that the spatial distribution of STH incidence is significantly associated with soil properties (sand proportion and pH), rainfall, wetlands and their uses, population density and proportion of rural residents. Identified spatial patterns are important for guiding STH prevention and control programs.

## 1. Introduction

Soil-transmitted helminths (STH) infect humans through their ingestion of parasite eggs (*Ascaris lumbricoides and Trichuris trichiura)* or via larvae that can penetrate the skin of humans (hookworm). STH infection mainly occurs in areas with moist and warm climates and poor sanitation and hygiene [1]. The burden of STH infections is well-known in Rwanda. After malaria, STH infections cause the highest morbidity and have considerable public health and economic impacts [2]. The countrywide mapping of STH conducted in 2008 and results from a simulation by the WHO in the same period [3] demonstrated that 27 out 30 districts in Rwanda have a STH prevalence above 50%. The 2008 prevalence mapping is based upon a cross-sectional survey at primary schools using the Kato–Katz technique [4]. Four or six schools are sampled per district, and data are aggregated to district level. The resulting STH prevalence per district guided subsequent massive drug administration (MDA) campaigns and other control efforts [5].

In 2005, the Rwandan Ministry of Health and its partners, including UNICEF, launched a biannual campaign called “Integrated Maternal and Child Health (MCH) Week”, including mass deworming with mebendazole for children aged 12 at 59 months, vitamin A supplementation for children aged 6 to 59 months and the distribution of family planning methods and other interventions. Since the establishment of the National Neglected Tropical Diseases (NTD) Program in mid-2007, efforts have been made to map and integrate NTDs (including STH) into the existing health system, including DHS2/HMIS, and to control them. In addition to children aged 12 to 59 months, biannual mass deworming targeting school aged children (5 to 15 years old) has been integrated into MCH week campaigns since 2008, after an initial mapping of STH (and schistosomiasis). Mebendazole or albendazole was used to treat soil-transmitted helminthiasis depending on the availability of donations [6].

The follow-up national remapping took place in 2014 [7,8]. The active monitoring of the spatial patterns of STH infection is needed to identify areas of high transmission, assess the impact of interventions and guide future control measures. Therefore, an efficient and inexpensive approach is needed that can be used to monitor the spatial distribution of STH transmission across space and over time.

The prevalence study conducted in 2008 has three main limitations. Firstly, the district-level mapping approach is too crude to capture the considerable geographic variability of biophysical and socio-economic conditions associated with STH transmission. Secondly, the existing STH prevalence inventory primarily targets school-aged children, thus not considering pre-school aged children, who are also much affected. Thirdly, differences in STHs pathogenicity and mode of transmission [9] are not considered. Prior research in countries neighbouring Rwanda has demonstrated that different species of STH infections have very different spatial patterns of transmission [10,11]. Additionally, Clements et al. [12] identified different spatial patterns for different STH species in the East African Great Lakes region, with ubiquitous hookworm infection and highly focal prevalence patterns of *A. lumbricoides* and *T. trichiura.*

STH infections are associated with biophysical features that influence egg and larval survival [13], and socio-economic factors such as inadequate sanitation, hygiene, and certain behavioural factors (e.g., defecation in the bush, reliance on traditional medicine) can facilitate transmission [14]. The latter may differ from one to another STH as they have different life cycles and transmission trajectories [15].

In Rwanda, health records on confirmed STH infections are routinely collected at the primary health facility level. When combined with small-area population data, the recorded case data can be used to estimate STH incidence rates for small geographic units, which we term Health Facility Service Areas (HFSAs). The approach used for delineating HFSAs is outlined in our earlier paper [16]. An important benefit of using incidence rates at the HFSA level is that detailed spatial patterns of STH transmission can be identified. We see the use of the HFSA-level incidence data as complementary to prevalence-based studies.

This research first investigates if incidence data can be used to identify the spatial distributions of STH transmission for individual helminth species as well as for combined STH infection. Second, it assesses how spatial variations in STH transmission are associated with environmental (physical and socio-economic) risk factors. Conducting such a geographically fine resolution assessment is novel as well as essential for a better understanding of the spatial variations in STH incidence in Rwanda and can contribute to geographically targeted future control efforts.

## 2. Materials and Methods

### 2.1. The Study Area and Health Facilities

Rwanda is characterized by diverse biophysical conditions, from lowland eastern savannas and hilly areas in the central-southern region to the western and north-western mountains. Rwanda has a temperate tropical highland climate, with little variation in the annual average temperature (20 °C). Rainfall varies with the topography and is characterized by with two rainy seasons.

Rwanda has around 11 million inhabitants, on average around 435 inhabitants per km^2^ [17], with a decentralized health facility structure. The primary health facility is the lowest level of the current health system hierarchy through which the health needs of the population are addressed [18]. Around 450 functioning primary health facilities are grouped into 367 HFSAs. (Figure 1). The approach used for delineating HFSA boundaries is explained in Nyandwi et al. [16].

### 2.2. Data Collection and Structuring

STH incidence and prevalence data. Confirmed cases of STH infection including roundworm (*A. lumbricoides*), whipworm (*T. trichiura*) and hookworm recorded at the primary health facility level for 2008 were obtained from the Rwanda Biomedical Centre—Malaria and Other Parasitic Diseases division (RBC/M&OPD). Prevalence data per district for individual and combined STH infections from the 2008 nationwide survey were also obtained from RBC/M&OPD. The year 2008 was selected for two reasons. First, it corresponds with the period of the nationwide school-based survey and mapping of STH prevalence (2008). Second, HFSAs can be accurately delineated given the widespread implementation of community-based health insurance, which determines which health facility the inhabitants of a given village or cell should attend when seeking care.

Population and general spatial data. Demographic data and socio-economic conditions per administrative sector for 2008 were obtained from the Integrated Household Living Conditions Surveys (EICV 2 & EICV 3) and the annual population growth rate reported by National Institute of Statistics of Rwanda. The following socio-economic variables were considered: (i) proportion of rural to urban area, (ii) population density [19], percentage of households with (un-) improved water source, (iii) percentage of households with access to improved sanitation, and (iv) education level (percentage of persons with six years basic education and persons with twelve years basic education).

General spatial data (e.g., lakes, islands, parks, roads) were obtained from the Centre for GIS and Remote Sensing of the University of Rwanda (CGIS-UR).

Environmental factors. Several variables were considered as potential risk factors for STH transmission (see Table 1). Three topographical parameters (elevation, slope and terrain shape index) were generated from a high-resolution digital terrain model produced from 2008 aerial photography [20,21]. Soil parameters (pH, clay and sand percentage) were extracted from the soil geo-database of Rwanda, the outcome of a semi-detailed soil survey of 1833 soil profiles spread over the country [22].

The annual averages of the climatic data (e.g., temperature, rainfall), measured at 183 weather stations from 1950 to 2010, were interpolated using the thin-plate smoothing spline algorithm as proposed by Hijmans et al. [23]. The boundaries of lakes, wetlands, and main rivers, produced by the Rwanda Environment Management Authority [24] and further adjusted by Nyandwi et al. [25], were also available for this study. For each HFSA, the following proxy factors were generated: (i) area and percentage of wetland and lake, (ii) wetland use (percentage of area covered by natural vegetation, (iii) total area of wetlands used for intensive irrigated agriculture).

**Table 1 tropicalmed-07-00202-t001:** Environmental risk factors used to assess the relationships with STH incidence.

Data Type	Variable	Source	Description
Topographic	Elevation, slope and Terrain Shape Index (TSI)	RNRA L&M	Using the DEM, an elevation map was created and further used to generated Slope and Terrain Shape Index [26] using ArcGIS. The outputs have a high spatial resolution with a cell size of 10 × 10 m.
Climatic	Rain, temperature humidity, evapo-transpiration	NMS	Data are measured nationwide at 183 meteorological and agro-meteorological weather stations and were interpolated and resampled to a cell size of 800 m.
Soil characteristics	pH, Clay and sand percentage	MINAGRI	The soil geo-database of Rwanda created in 2000 by MINAGRI in collaboration with Ghent University, Belgium. Resolution: 500 m (1/50,000)
Water bodies & Wetlands	Wetland area, wetland proportion, wetland use	REMA & RNRA	National inventory of wetlands by REMA in 2008 updated in collaboration with RNRA in 2012. Ortho-photograph resolution: 0.25 m.
	Rice crop area	MINAGRI	The GIS database of the Rural Sector Support Program of MINAGRI considering new wetland management or rehabilitation for intensive cropping. Resolution: 25 m.
Socio-economic conditions	Demography: Population, pop. density, number of households	NISR	The fourth Rwandan Population and Housing Census, August 2012
	Residential area		
	Sanitation		
	Water source		
	Education level		

Datasets were obtained from: Rwanda Natural Resource Authority, Land and Mapping Department (RNRA/L&M), Ministry of Agriculture and Animal Husbandry (MINAGRI), Rwanda Environmental Management Authority (REMA), National Institute of Statistics of Rwanda (NISR).

### 2.3. Comparison of Spatial Patterns of STH Incidence and Prevalence at District and HFSA Level

Incidence and prevalence are both measurements of disease frequency. Incidence estimates how often disease occurs in a given area and time period (a measure of disease risk). Prevalence evaluates how a disease is spread in a given population (a measure of disease burden) at a given moment in time. The first step of our analysis evaluates if areas with high STH prevalence correspond with areas with high STH incidence, for which we compare the 2008 prevalence at each of the 136 surveyed schools with incidence rates at the corresponding HFSA. Both prevalence and incidence STH data were also aggregated and mapped at district level to enable visual comparison.

### 2.4. Converting Confirmed Cases of STH Infection to Incidence Rates

The number of confirmed STH cases per HFSA per month was summed per year and subsequently aggregated to the district level. The annual number of confirmed cases and demographic data were combined using Excel and then linked to the district and HFSA boundary data used for visualization. The incidence rate is computed as the number of cases per district (*n* = 30) or HFSA (*n* = 367) divided by the population of that district or HFSA, as computed with the equation below:Di = (In/Pt) × 1000 (0) *
where Di is the incidence rate, In is the total number of new cases in 12 months of a year per district/HFSA and Pt is the total population of that year for that entity. * Rates are reported per 10,000 persons at the district level and per 1000 persons at the HFSA level [27].

To account for rate instability, raw incidence rates of sparsely populated HFSAs were replaced with weighted averages using empirical Bayesian smoothing (EBS) [28]. Data were smoothed using SpaceStat software [29]. For the visualization, incidence rates were classified into four classes using the Jenks classification. The same classes were compared across years and between incidence and prevalence maps at the district level. Furthermore, the average incidence rates for 2008 at the HFSA level were superimposed on the points map of the 136 schools surveyed during the nationwide mapping of 2008. A scatter plot was then generated to illustrate the association between incidence rates at the HFSA level and prevalence rates per school.

### 2.5. Spatial Autocorrelation Analysis and Test for Spatial Clustering

Spatial autocorrelation was computed to ascertain if spatial clustering occurs between neighbouring HFSAs. We conducted an exploratory data analysis by plotting the semivariogram and conducting the Moran’s index test [30]. Moran’s I was used to identify spatial clusters using cluster and outlier analysis. Negative values indicate negative spatial autocorrelation; positive values indicate the reverse. Values range from −1 (indicating perfect dispersion) to +1 (perfect correlation). A zero value indicates a random spatial pattern [31].

### 2.6. Assessment of Relationship between STHs Incidence and Environmental Covariates

Incidence data and potentially associated environmental factor data were integrated at the HFSA level using ArcGIS 10.2.2. All bio-physical factors maps were standardized to a resolution of 10 × 10 m. Average values for potential explanatory variables for each District and HFSA were extracted using the zonal statistic module of ArcGIS 10.2.2. After that, all data were exported to IBM SPSS Statistics, version 20 for further analysis.

The choice of explanatory variables is an important step when constructing any statistical model [32]. Before training the model, the selected variables were screened for collinearity [33]. The next step was to split the available HFSA-level data (*n* = 367) into calibration and validation subsets. We randomly selected 60% of the available data as training the dataset (*n* = 220) for model fitting and the remaining 40% for model validation (*n* = 147). The proportions of the two subsets follow the principle of using more data for calibration, and a validation subset reaching at least 35% [34,35]. The validation and calibration data were selected using an online list randomizer tool (https://www.random.org/lists/ (accessed on 15 January 2016).

EBS-smoothed STH incidence rates, as dependent variable *y* were related to the potentially explanatory variables as variables *X*_1_ to *X_n_*. The purpose was to quantify the strength of the relationship between STH incidence and the significant covariates [36,37]. We also report the standardized coefficient to identify which of the independent variables have a greater effect on the dependent variable. This is needed since our variables have different units of measurement [38].

## 3. Results

### 3.1. Spatial Patterns of STH Incidence at District Level

The total number of confirmed STH infections was 1,204,330 for 2008. At the district level, the number of reported cases ranged from 22,000 to 78,000. If we consider individual STH species, very significant differences can be observed (see Figure 2). Rwandans are frequently infected with *T. trichiura*, amounting to 57% of all recorded STH cases, followed by *A. lumbricoides* (38%) and a much lower infection rate with Hookworm (5%). The district-level incidence of individual STH species and of co-infection for the population at risk is summarized in Figure 2 and visualized in Figure 3a–d.

Some Districts (e.g., Rwamagana ad Ngororero) have relatively low incidence rates. Others such as, for example, Musanze and Huye exhibit high incidence rates.

In Figure 2, combined STH infections show incidence rates ranging between 140 and 800 per 10,000 persons (Figure 3a). *T. trichiura* is a nationwide problem in Rwanda, with incidence rates ranging between 78 and 500 per 10,000 persons (Figure 3b). *Ascaris lumbricoides* incidence follows with rates ranging between 15 and 410 per 10,000 persons (Figure 3c). Hookworm incidence rates are much lower, varying between 3 and 65 per 10,000 persons (Figure 3d). The Northern Province is the most infected region. Kicukiro District (of Kigali city, Rwanda) exhibits a high incidence rate in general and specifically for *T. trichiura* (500 per 10,000 persons).

### 3.2. Comparing District Level Prevalence and Incidence Data

To check if there is a significant relationship between prevalence and incidence data, we compared district-level incidence rates from 2008 with district-level prevalence data based on the 2008 nationwide school-based survey. Figure 3 illustrates both incidence and prevalence data at district level for individual STH species and for all infections combined, as well as scatter plots that illustrate the correlation between the two. We can observe that the correlations vary considerably, with correlation coefficients of 0.63, 0.27, 0.1 and 0.02 for *A. lumbricoides*, Hookworm, combined STH infection and *T. trichiura*, respectively. The high correlation coefficient for *A. lumbricoides* indicates that routinely recorded cases of *A. lumbricoides* can be used to complement prevalence data extrapolated to the district level. To a much lesser degree, this also is the case for hookworm. For combined STH infections and *T. trichiura*, the prevalence and incidence rates are very different.

### 3.3. Spatial Patterns and Spatial Clusters of STH Transmission at the HFSA Level

In the following, we further analyse the spatial patterns and clustering of STH transmission at the detailed spatial scale of the HFSA, as illustrated in Figure 4.

Figure 4a shows that combined STH infection is widespread in Rwanda, with medium to high incidence rates in many HFSAs throughout the country. The spatial patterns of *T. trichiura* are overall similar, as visualized in Figure 4b. *A. lumbricoides* on the other hand exhibits a much more focalized spatial distribution with areas of high transmission in the northern, southern and south-western areas of the country (see Figure 4c). As illustrated in Figure 4d, only a few HFSAs have high transmission rates of hookworm (namely, Kabuga HFSA in Kamonyi district, Gatare HFSA in Nyamasheke district, Rwahi HFSA in Rulindo district, Rutenderi HFSA in Gakenke district and Mucaca HFSA in Burera district). Apart from these areas, hookworm does not have high incidence rates in Rwanda.

Spatial auto-correlation analysis was subsequently performed per STH species and for combined STH incidence at the HFSA level using Moran’s index. Outcomes are illustrated in Table 2 and show positive and statistically significant spatial autocorrelations.

Moran’s I statistic indicates that there is a less than 1% likelihood that the clustered pattern could be the result of random chance for *A. lumbricoides* and for combined STH infections. The spatial clustering of *T. trichiura* has a less than 5% likelihood of being the result of random chance. The spatial pattern of hookworm is random.

The Anselin local Moran’s I cluster analysis results at the HFSA level presents three levels of spatial association (see Figure 5): no significant association, positive spatial association (the High–High and Low–Low clusters) and negative spatial association (the High–Low and Low–High outliers).

### 3.4. STH Incidence at District and HFSA Level and Their Association with Environmental Factors

The best-fitting model was obtained using a stepwise linear regression with a random distribution of residuals producing the highest R-squared and lowest standard error. The environmental factors associated with STH incidences vary per STH species and with the spatial scale of analysis. Table 3 summarizes the statistical outputs of correlated ecological factors (low elevation, less sandy soil, percentage of area covered by wetland, wetland use for rice cultivation), climatic conditions (rainfall) and socio-demographic characteristics (rural population, number of households, poor sanitary conditions).

The exploratory regression models indicate that environmental factors explain between 13 and 30% of the total variability in *T. trichiura*, *A. lumbricoides* and combined STH cases at different spatial scale. The combined STH incidence model reached 82% at the district level. Hookworm incidence was poorly explained by the available risk factors, reaching only 8% at the HFSA level (see Table 4).

## 4. Discussion

These findings illustrated that incidence rates of especially *T. trichuira* were much higher than that of hookworm in 2008 (see Figure 2). Most likely, this is related to the fact that *T. trichuira* is highly unresponsive to a single regimen of albendazole and mebendazole [39]. We also demonstrated that the spatial distribution of STH incidence is quite different among individual STH species. Thus, it is advisable to separately assess spatial patterns of STH incidence for each STH species. The readily available systematically recorded routine health data at detailed spatial scale makes the spatial assessment of individual STHs simple and cheaper. By using data of individual STH species, we can separately also assess the relationships between potentially associated risk factors and STH incidence more directly. Although STHs are considered to be widespread in Rwanda, the distinctive local characteristics and the association with biophysical parameters (such as wetlands and wetland use) is significant.

*Incidence rates versus prevalence data at the district level.* We observed a clear relationship between the incidence and prevalence data at the district level for some but not all STH species. There is a significant positive correlation between incidence and prevalence data for *A. lumbricoides* and hookworm, while for *T. trichiura* and the STH combined, only a weak positive significant correlation was observed. The prevalence data are extrapolated to the district level on the basis of results obtained at four to six surveyed schools per district. The Rwandan cross-sectional prevalence survey follows the World Health Organization guidelines [40], but it has some limitations. One limitation is that the sampling strategy used is not spatially stratified, and it includes only a limited number of schools to represent a large geographical area with internally varying risk of infection [12]. In addition, the primary school-based prevalence surveys focus on school age children, while pre-school children might be much more exposed. In reality, the increase of wetland reclamation for rice cultivation during the last decade [41] has most probably increased exposure for adults involved in rice cropping activities (i.e., women and their infants).

*Different spatial patterns for individual STH species.* Routinely collected data captured at the health facility level can be used to identify and visualize the spatial patterns of STH incidence at a fine geographic resolution, as illustrated in Figure 2. STH transmission is not uniformly distributed across the country. The presence of spatial clusters, detected by Moran’s I and LISA (Table 2 and Figure 5), confirms this. Previous studies have also reported on the disparate spatial distributions of different STH species [42]. This study has highlighted that *T. trichiura* is geographically widespread in Rwanda, with 57% of the population infected. More importantly, the spatial patterns of the three STH species are very different. This is in line with Clements et al. [12], who also identified different spatial patterns for the same STH species in Uganda, Tanzania, Kenya and Burundi. There are relatively few cases of hookworm compared with other intestinal helminths in Rwanda. Since transmission to humans usually occurs through bare feet on contaminated soil, the limited number of reported cases may be linked to improved socio-economic conditions in Rwanda during the last decade [43], leading to the increased wearing of shoes.

*STH species incidence is associated with different environmental variables*. The regression models demonstrate different fits at district and HFSA level and under different environmental conditions. The multiple regressions generated models with four independent contributions: from soil sand content, rainfall, area proportion of wetland and demography. It appears that in general, the STHs are more likely to infect people in more populated rural areas with less sandy soil and higher rainfall and in wetland-rich areas. The differences between associated factors and STH incidence appear to be related to different helminth living conditions and different transmission mechanisms, which vary geographically [15]. *A. lumbricoides* incidence is mainly associated with densely populated rural areas with less sandy soils and high rainfall. *T. trichiura* incidence, on the other hand, is clearly associated with wetlands outside urban areas, while Hookworm is weakly associated with lower altitudes.

Many of the associated biophysical factors are directly or indirectly related to wetlands environments. During the last decade, rice has become a major food crop in Rwanda; as a result, the area for rice production increased from 4000 to 16,000 hectares between 2000 and 2010 [41]. Overall, our findings are consistent with other studies [12,44,45], who found that soil moisture and relative atmospheric humidity are important environmental determinants of STH development and transmission.

The relationship between high population density and intestinal helminth incidence comes from residential settings allowing favourable conditions for transmission [46]. In Rwanda, with persistent and growing urban and sub-urban agriculture [47], this remains a problem. Again, our findings are congruent with other studies that identify the use of unimproved water sources for domestic use and poor sanitary and housing conditions as important risk factors. Mascarini-Serra ([45], p. 178) nicely captures this by stating, “The prevalence of STHs in the community can be used as an indicator of the conditions of living, environmental sanitation, level of education, and the socio-economic status of the community”.

*Limitations of the study.* The number of confirmed cases recorded at health facilities may be only a fraction of all infected persons for a number of reasons. First, self-treatment is common in Rwanda, as it is in many developing countries [48]. Often families practice regular de-worming using either traditional medicinal treatment or anti-intestinal medication obtained from a pharmacy. Second, reported cases will probably involve patients with more severe infections and with clear clinical symptoms. Third, the routine health data used for this study are collected from public and faith-based health facilities only. Since the proportion of people seeking private health care in Rwanda is limited, this is considered to only have a minor influence [49]. Fourth, the proportion of reported cases may also be influenced by variations in health-seeking behaviour. This may explain the relatively high number of confirmed STH cases in urbanized areas (i.e., Kicukiro district of Kigali city and Huye district) inhabited by relatively well-educated population groups. On the positive side, the Rwandan Health Management Information System systematically and accurately records laboratory confirmed STH cases at a fine geographic resolution [50]. The setup of the community-based health insurance furthermore makes it possible to accurately delineate the geographic service area of health facilities. Because of this, we feel that recorded STH cases can be adequately used identify and monitor the spatial and temporal patterns of STH transmission for individual STH species in Rwanda.

## 5. Conclusions

Routinely collected confirmed STH cases are suitable for generating spatially explicit overviews of the distribution and intensity of STH transmission for small geographic units, which we here termed HFSA. The distributions of individual and combined STH incidence in Rwanda varies per helminth species and with environmental conditions. It is also clear that the STH occurrence is geographically clustered in areas much smaller than the district level, which has been used to date. *T. trichiura* is highly endemic, with a broad spatial distribution throughout the country. Hookworm has the lowest incidence rates with an almost random spatial distribution.

A significant geographical association is observed between STH incidence and specific biophysical and socio-demographic factors. The most strongly associated biophysical factors are directly or indirectly related to wetland environments. The number of STH cases is also associated with increasing rainfall, population density, the proportion of the rural population, the area covered by wetlands and rice cropping and less sandy soils and at lowest elevation levels.

Since the routine health data used in this study are readily available, the approach described in this study can be used to design the spatial sampling for prevalence and/or to complement the prevalence data. Finally, the presented methodology can contribute to more effective, location-specific STH control program design and evaluation.

## Figures and Tables

**Figure 1 tropicalmed-07-00202-f001:**
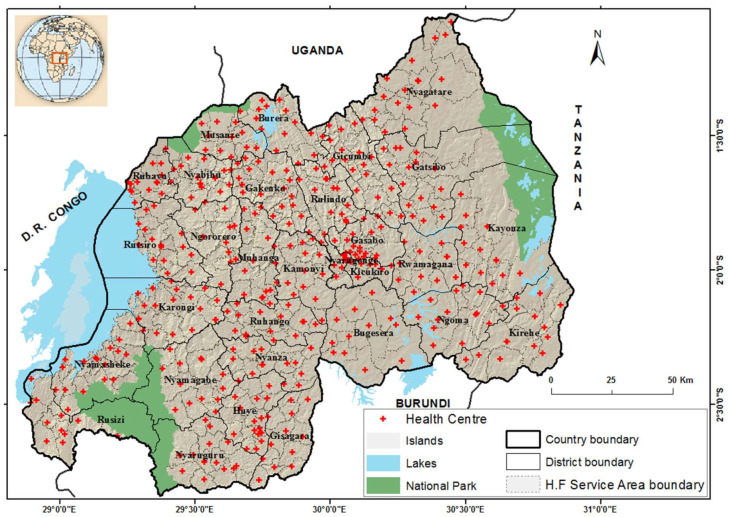
District boundaries, primary health facility locations as provided by RBC–GIS unit and Health Facility Service Area (HFSA) boundaries delineated using spatial allocation tools and data from the community-based health insurance scheme in 2010.

**Figure 2 tropicalmed-07-00202-f002:**
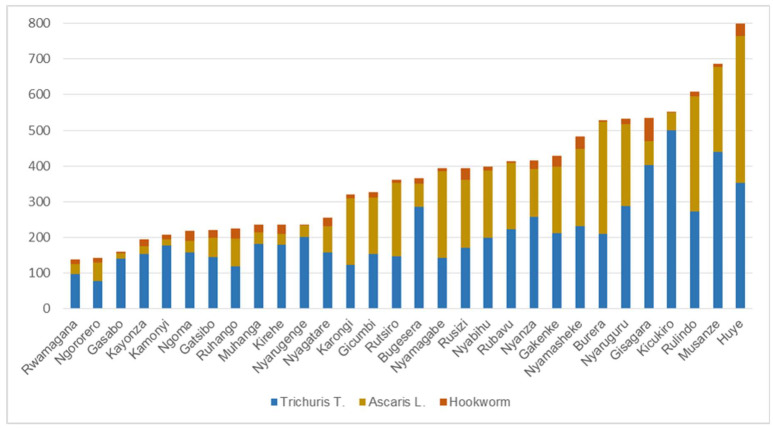
STH incidence rates (per 10,000 persons) per district in Rwanda for 2008, as reported in RHIMS.

**Figure 3 tropicalmed-07-00202-f003:**
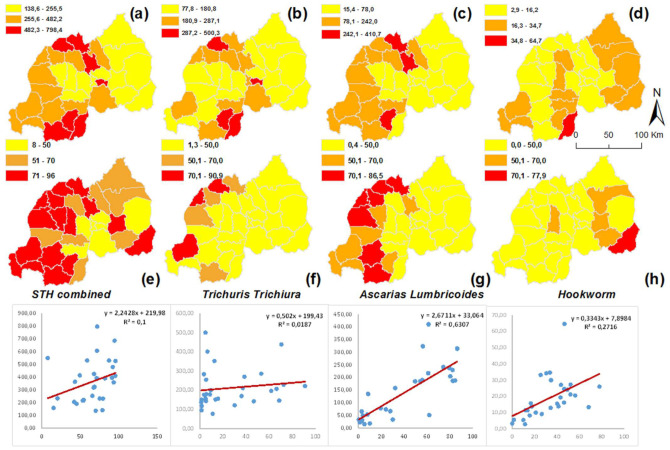
STH incidence and prevalence rates and their correlation at the district level. Starting from the left-upper corner, incidence maps for combined STHs (**a**), for *T. trichiura* (**b**), for *A. lumbricoides*, (**c**) and for hookworm (**d**). The prevalence maps and scatter plots are shown for combined STH cases (**e**), individual cases with *T. trichiura* (**f**), *A. lumbricoides* (**g**) and hookworm (**h**).

**Figure 4 tropicalmed-07-00202-f004:**
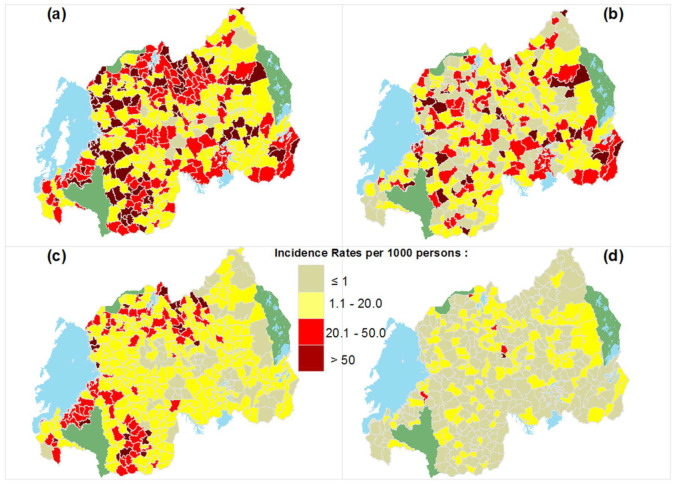
Incidence rates at HFSA level for combined STH infection (**a**), *T. trichiura* (**b**), *A. lumbricoides* (**c**) and hookworm (**d**).

**Figure 5 tropicalmed-07-00202-f005:**
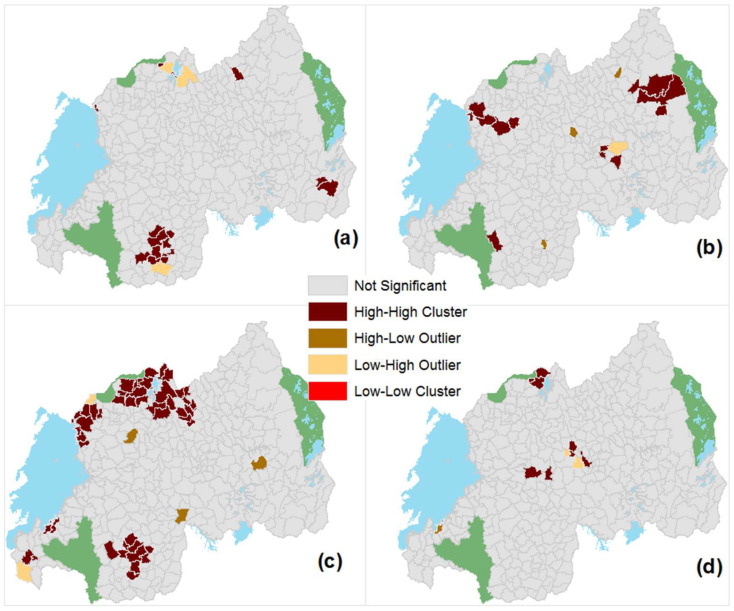
Spatial distributions of Anselin local Moran’s I at the HFSA level, for (**a**) combined STH cases; (**b**) *T. trichiura*; (**c**) *A. lumbricoides* and (**d**) hookworm.

**Table 2 tropicalmed-07-00202-t002:** Moran’s index computed for each STH species and for and combined STH incidence rates at the HFSA level.

Year	Moran’s I	*p*-Value	z-Score
Combined STH	0.113	0.000006	4.51
*Trichuris trichiura*	0.052	0.037	2.08
*Ascaris lumbricoides*	0.375	0.000000	14.433
Hookworm	0.0056	0.744	0.326

**Table 3 tropicalmed-07-00202-t003:** Standardized beta coefficients of associated factors at District and HFSA spatial scale.

Factors	Combined		*Trichuris Trichiura*		*Ascaris Lumbricoides*		Hookworm	
	District	HFSA	District	HFSA	District	HFSA	*district*	*hfsa*
pH	-	-	0.43 **	-	-	-	-	-
Sand prop (%)	−0.38 **	−0.41 **	-	-	-	−0.43 **	-	-
Elevation (m)	−0.53 **	-	-	-	-	-	-	−0.29 **
Wetland area (ha)	-	-	-	-	-	-	-	-
Wetland proportion (%)	0.39 **	0.22 **	-	0.46 **	-	-	-	-
Wetland rice cultivation (ha)	-	-	-	0.26 *	-	-	-	-
Rainfall	0.43 **	0.29 **	-	-	-	0.31 **	-	-
Number of households	-	-	-	-	0.37 *	-	-	-
Unimproved sanitation	-	-	0.37 *	-	-	-	-	-
Rural	-	0.125 *	-	-	-	0.12 *	-	-
Urban	-	-	-	−0.27 *	-	-	-	-

Coefficient presented are significant with: *p* < 0.05 (*) and *p* < 0.01 (**).

**Table 4 tropicalmed-07-00202-t004:** Summary of linear regression models for incidence rates per STH species.

Spatial Level	Factor	Significant Variables	R^2^	Std. Error of the Estimate
*Ascaris lumbricoides*				
HFSA	Climatic	Rain	0.27	337.802
	Physical	Sand percentage		
	Socio-economic	Percentage rural population		
District	Demographic	Number of households	0.13	2952.698
Hookworm				
HFSA	Physical	Elevation	0.08	52.635
District				
*Trichuris trichiura*				
HFSA	Ecological	Wetland cultivated area	0.22	90.302
		Wetland proportion		
	Demographic	Urban proportion		
District	Physical	pH	0.30	395.764
	Demographic	Unimproved sanitation		
*Combined STH*				
HFSA	Climatic	Rain	0.24	362.725
	Physical	Sand percentage		
	Ecological	Wetland prop		
	Demographic	Rural proportion		
District	Physical	Elevation, Sand percentage	0.823	1553.642

## Data Availability

Most of the data presented in this study are available on request from the corresponding author. STH infection data for 2008 were obtained from the Rwanda Biomedical Centre—Malaria and Other Parasitic Diseases division and are available from the corresponding author only with the permission of the Rwanda Biomedical Centre—Malaria and Other Parasitic Diseases division.

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
