# Peer review of "Using Routinely Collected Health Records to Identify the Fine-Resolution Spatial Patterns of Soil-Transmitted Helminth Infections in Rwanda"

_tropicalmed, 2022, doi:10.3390/tropicalmed7080202_

Round 1
Reviewer 1 Report
In the manuscript "Using routinely collected health records to identify fine resolution spatial patterns of soil-transmitted helminth infections in Rwanda", they present and interesing and useful methodology to determine the spatial distribution of the STH species present and their prevalence and incidence at the national scale using data collected by the health system.
I believe the study is an important contribution and have only minor comments for the authors:
Line 39-40 in the introduction, a clarification is needed since it would seem that hookworm is also acquired orally instead of penetration of larvae through the skin.
Please verify that all the species names are in italics. Also, there is no need to write out the species name extensively every time. When you mention the species first you write it out completely and then abbreviate it, i.e. Ascaris lumbricoides, A. lumbricoides. Except when a species name is used to start a sentence, then, it needs to be written out completly.
Having a bit of background information on the deworming program of Rwanda, when did it start, was it continuous, with what anthelminthic (ALB or MEB), how often (once or twice a year) and also, what techniques were used to determine the prevalence (was it KK?), would help to better interpret the results. Also, if KK was used, the intensity of infection should be used as a variable since that will give you a measure of how effective the deworming program is and allows you to detect high transmission areas. In previous studies, certain environmental variables have been associated with presence of STH but socioeconomic and behavioral variables were more associated with infection intensity. This is important data to be able to discuss in the discussion section, since it´s not clear to me if the data used is from 2007-2008, what is the situation today. Is the deworming program still ongoing until today or was it interrupted?
Usually, tables and figures need to be able to stand alone. Therefore, the table and figure legends should contain all the relevant information needed to interpet them. In the figure legends, it would be helpful to include the data information (years and locality). In Table 3, species names should be written out completely and in the correct order, i.e. Trichuris trichiura.
In the discussion, with respectt to the high prevalence of T. trichuira, it should be mentioned that both albendazole and mebendazole are not very effective agains this STH, and this is why you are probalby seeing a reduction in A. lumbricoides and hookworm, but not so much in T. trichiura. There are several references in the literature on this topic and the need for new anthelminthics of combination of anthelminthics. With respect to hookworm, usually the prevalence of this STH is increases with age; therefore, this should be taken into consideration when interpreting the results, given that lymphatic filariasis is not endemic in the country and therefore community deworming programs are not implemented.
The discussion should take into consideration other studies that have explored the association between STH prevalence and environmental/socioeconomic/behavioral characteristics to be able to compare and contrast results.
Verify the format of the references.
Reviewer 2 Report
Up-to-date information related, Literature review did not mention about STH risk factors, Some reference did not consistent with the claim in content.
1. Row 121 Reference 18, In the reference refer to several tropical diseases except STH, The factors in the reference is inconsistent with the content article.
Row 120-122 The following socio-economic var- 120 iables were considered: (i) proportion of rural to urban area, (ii) population density, [18] 121 percentage of households with (un-) improved water source
Reference 18
Interpretation
Population growth, increased average age of the world's population, and largely decreasing age-specific, sex-specific, and cause-specific death rates combine to drive a broad shift from communicable, maternal, neonatal, and nutritional causes towards non-communicable diseases. Nevertheless, communicable, maternal, neonatal, and nutritional causes remain the dominant causes of YLLs in sub-Saharan Africa. Overlaid on this general pattern of the epidemiological transition, marked regional variation exists in many causes, such as interpersonal violence, suicide, liver cancer, diabetes, cirrhosis, Chagas disease, African trypanosomiasis, melanoma, and others. Regional heterogeneity highlights the importance of sound epidemiological assessments of the causes of death on a regular basis.
2. The information really outdates 2007-2008.
3. The information used to comparative reference is different in each year and is outdate
4. Reference 1; no edible plants

Round 2
Reviewer 2 Report
No